# LogBD: A Log Anomaly Detection Method Based on Pretrained Models and Domain Adaptation

**Shuxian Liu \*, Le Deng, Huan Xu and Wei Wang** 

College of Information Science and Engineering, Xinjiang University, Urumqi 830046, China;
sc_dengle@stu.xju.edu.cn (L.D.); xhuan@stu.xju.edu.cn (H.X.); wangwei33@stu.xju.edu.cn (W.W.)
\* Correspondence: liushuxian@xju.edu.cn

**Abstract:** The log data generated during operation of a software system contain information about the system, and using logs for anomaly detection can detect system failures in a timely manner. Most existing log anomaly detection methods are specific to a particular system, have cold-start problems, and are sensitive to updates in log format. In this paper, we propose a log anomaly detection method LogBD based on pretrained models and domain adaptation, which uses the pretraining model BERT to learn the semantic information of logs. This method can solve problems caused by the multiple meaning of words and log statement updates. The distance to determine anomalies in LogBD is constructed on the basis of domain adaptation, using TCNs to extract common features of different system logs and mapping them to the same hypersphere space. Lastly, experiments were conducted on two publicly available datasets to evaluate the method. The experimental results showed that the method can better solve the log instability problem and exhibits some improvement in the cross-system log anomaly detection effect.

**Keywords:** log anomaly detection; pretrained models; domain adaptation

## 1. Introduction

With the development of society, computer software is becoming more widely used in various fields of life, and the scale of software is increasing [1]. Accordingly, the technologies and configurations involved in software systems usually evolve, and their hardware and software inevitably fail. In the wake of the Newcastle pneumonia outbreak, industries worldwide accelerated their shift to online services, and online services in some key industries have profoundly affected the daily lives of people in various countries. The downtime of software systems that provide critical services may have a huge impact on society.

The increasing user base and gradually diversifying functional requirements have led to the expansion of the functional scope of various information systems, the gradual complexity of system architectures, and the existence of a large number of many potential risks, resulting in frequent incidents. In 2015, Amazon Web Services (AWS) experienced a massive downtime that lasted more than 40 s [2]. This resulted in unresponsiveness for several apps such as Slack, Asana, Netflix, Pinterest, and multiple websites using AWS services. In 2018, the AWS management console failed for nearly 6 h, threatening its sales activities for 36 h [2]. In 2020, numerous services and major applications of Google, such as YouTube and Gmail, were inaccessible, and millions of users were affected to varying degrees with outage conditions [2]. In 2021, several key US federal-operated payment systems, which underpin millions of financial transactions, suffered service disruptions due to operational errors [2]. This service disruption, in turn, was caused by a failure within the Federal Reserve's systems. Therefore, after a system or software is up and running, it is of paramount importance to maintain its proper functioning and to detect and deal with failures in a timely manner.

Maintaining and ensuring the availability of the entire system are the responsibilities of the operation and maintenance staff. In the face of increasing software reliability requirements, early operation and maintenance work is carried out manually. In the era of rapid expansion of Internet businesses and rising labor costs, traditional manual operation and maintenance have been unsustainable and cannot keep up with the rapid development of the Internet [3]. Therefore, automated operation and maintenance activities are essential. This approach mainly uses scripts that can predefine rules and automatically trigger [4] routine and repetitive operation and maintenance tasks, thereby reducing operation and maintenance costs and improving operation and maintenance efficiency. Automatic operation and maintenance activities utilize expert systems based on industry, as well as knowledge of the operation and maintenance scene. With the rapid growth of Internet business scope and the diversification of service types, expert systems based on manually specified rules cannot meet the demand. The emergence of DevOps partly solves this problem, but it pays more attention to the global perspective. Artificial intelligent operation and maintenance (AIOps) is an advanced implementation of DevOps in operation and maintenance. The concept of AIOps was proposed by Gartner in 2016 [5]. The goal of AIOps is to solve the problems of inefficiency, poor accuracy, and lack of automation in traditional IT operations. It is based on data management, algorithm, and scene-driven elements, combined with big data and machine learning technology. Data analysis in AIOps is no longer limited to the scope of the data itself, but includes mining specific patterns to predict possible events or diagnose the root cause of the current system behavior. AIOps enables operators to take actions and solve problems more intelligently. One of the focuses of AIOps is fault detection [6]. Logs play an indispensable role in fault detection.

Log files are among the primary methods for recording operational status in IT domains such as operating systems. They are an important resource for identifying whether a system is in a healthy state, containing a wealth of system information. System developers and operations and maintenance personnel often use log data to view system status, find system failures, and locate root causes. The rich information hidden provides a good perspective for analyzing software system problems. Therefore, mining log information in large amounts of log data can help enhance system health, stability, and availability, and help operations and maintenance personnel to find abnormalities in system operation status.

Most of the existing approaches learn normal patterns from many normal logs, and the models examine the abnormal log sequences on the basis of the learned normal patterns. Most of the methods use static word embedding methods, such as Word2Vec or Glove, but words may have different meanings in different contexts, and dynamic word embedding methods are needed. The above methods are also not studied for cross-system log anomaly detection; they are all for a specific system. The models obtained from training, therefore, do not work well on new systems. In this paper, we propose LogBD to build a cross-system log anomaly detection model to detect anomalies in both source and target systems, using the semantic and sequential relationships of log messages to achieve anomaly detection results with very few logs in the target system.

The main contributions of this paper are as follows:

1. BERT [7] is used to extract the semantic features of log messages and feed the log sequence vector with preserved semantic information into a neural network, which learns the normal patterns and semantic information of logs and is better able detect abnormal log sequences.
2. The log sequence vector is input into a temporal convolutional network (TCN) [8] for data mapping, and the hypersphere objective function is used to map the log sequences into the hypersphere space. The distance between normal log sequences and abnormal log sequences to the center of the sphere differs as a function of the distance mapped to the center of the sphere, enabling us to determine anomalies.
3. Domain adaptation learning is used to train the TCN so that it can extract the common features of log data from different systems and apply the "knowledge" learned from

the source system data directly to the target system data, thus enabling the detection model to detect anomalies from multiple systems.

The remainder of this paper is organized as follows: the relevant knowledge and related work are presented in Section 2; the details of our study are presented in Section 3; the performance of the proposed model is evaluated in Section 4; a summary of this work is provided in Section 5.

## 2. Related Work

Log anomaly detection generally has four steps, focusing on three parts: log parsing, feature selection, and anomaly detection. We address these three parts to introduce the related work.

### 2.1. Log Parsing

Log parsing is the task of breaking the raw log statements into log templates or log events. The body of a log statement usually consists of two parts: (1) templates are static keywords describing system events, usually a piece of natural language, that are explicitly written in the code of the log statement; (2) parameters, also called dynamic variables, are the values of a variable during the program runtime. In logging exception detection, the log content is the most important. Log parsing is generally applied to parse the log content, replace the variables, and keep the constants to get the log template, as shown in Figure 1.

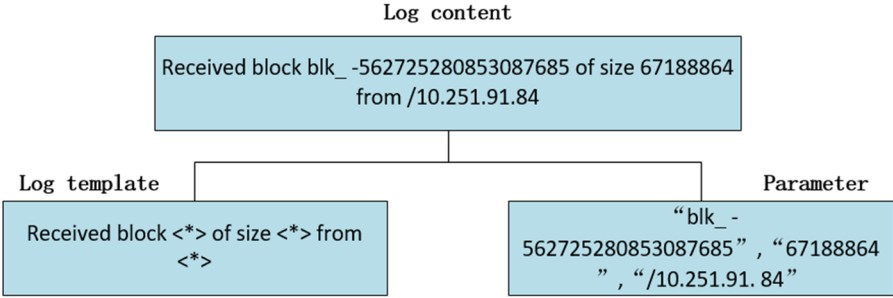

**Figure 1.** Sample of log parsing.

Log parsing has been studied by many scholars, and log parsing algorithms are divided into three main categories: clustering-based log pattern parsing algorithms, frequent item mining-based log pattern parsing algorithms, and heuristic-based log pattern parsing algorithms. The main clustering-based log pattern parsing algorithm is the Spell method [9], which is a structured streaming event log parser based on longest common subsequence (LCS) to parse unstructured log messages into structured log types and parameters in an online streaming fashion. The main log pattern parsing algorithm based on frequent item mining is the FT-Tree [10] algorithm, which is a general log template generation method that uses an extended prefix tree structure to represent switch syslog message templates, where the subtypes of the detail fields are usually made up of the longest combinations of frequently occurring words. Extracting the templates is equivalent to identifying from the syslog messages the longest combination of frequently occurring words from syslog messages. The main heuristic-based log pattern parsing algorithm is the Drain [11] algorithm, which is a fixed-depth tree-based approach that enables streaming and timely log parsing. Drain uses domain knowledge and regular expressions to preprocess new logs as they arrive. Drain then searches for log groups according to the encoding rules of the nodes inside the tree. If a suitable log group is found, the log message is matched to a log event stored in that log group. Otherwise, a new log group is created on the basis of log messages.

## 2.2. Feature Extraction

After processing the raw logs and before anomaly detection, log feature representation is also needed because log templates in the form of text data cannot be processed directly by computers. Feature representation is needed before model training and anomaly detection to learn normal or abnormal patterns of logs. In previous studies, scholars selected features from system logs [12], including log templates, number of events, intervals, event indices, and log variables, and then encoded them for use in neural networks by unique thermal coding or other methods. For example, event sequences are represented as event ID sequence vectors (IDs), i.e., sequences of log templates in chronological order, and they are fed into a recurrent neural network (RNN) to learn temporal correlations, treating unusual template sequences among them as anomalies. One-hot encoding is one of the most common techniques for processing categorical data and is, thus, often applied to encoding event types or tokens. Most scholars use one-hot encoding data directly as input to neural networks, but it can also be combined with other features (e.g., counting vectors) so that neural networks can recognize the input and learn different models for different log templates. One-hot encoding may be two-dimensional in some cases where the amount of data is too large. Some scholars use NLP methods as a replacement, such as Word2Vec and Glove, which are able to learn the semantic information of logs and better utilize the information contained in log data.

## 2.3. Anomaly Detection

The existing anomaly detection methods based on log data are mainly rule-based, graph-based, and machine learning-based detection methods [12]. Rule-based log anomaly detection methods use the frequency of keywords appearing in logs, combine domain knowledge, and design relevant regular expressions to mine potential information in log data. Graph-based anomaly detection uses the contextual relationships of log sequences to model and construct a request execution logic graph. Currently, machine learning-based methods include traditional machine learning methods and deep learning methods, which mainly use neural network models to infer whether logs are abnormal by judging log sequence relationships.

Among the traditional machine learning methods, Xu et al., constructed a log anomaly detection method based on principal component analysis (PCA) [13], which constructs log feature vectors from the state ratio vectors and event count vectors during system execution, and combines PCA with term weighting techniques in the field of data retrieval, which can discover patterns between event count vectors. Chen et al., proposed a decision tree based detection method [14], which uses decision trees to handle discrete and continuous attributes to perform anomaly detection by analyzing server request sequences. Meng proposed LogClass [15], which uses a data-driven approach to detect and classify anomalies in logs. First, the raw log data are preprocessed to generate a bag-of-words model description approach. Then, positive and unlabeled (PU) learning methods and support vector machines are used to train anomaly detection and classification models. Lastly, anomaly detection is performed on logs. Lou et al., proposed invariant mining (IM) to mine information in log data [16], which mines invariants from console logs using a strong search algorithm, and then detects system anomalies if the appearance of a new log sequence breaks an invariant during the life cycle of system execution. Vaarandi et al., proposed logcluster [17], a log clustering-based problem identification method that uses hierarchical clustering techniques to aggregate similar log sequences into clusters for automatic problem identification.

As the size of log data increases, more human and material resources are required for feature engineering, and the cost of traditional machine learning methods increases greatly. Many scholars have applied deep learning methods to log anomaly detection. Du et al., proposed DeepLog [18], a deep learning model based on a long short-term memory network (LSTM), which was the first to use deep learning for detecting log data sequences that constructs log sequences as time series to achieve the anomaly detection of log template

sequences and parameter values. Meng et al., designed a vectorized representation of templates inspired by word embedding, template2Vec; they proposed LogAnomaly [19], an anomaly detection framework for unstructured log streams. Zhang et al., proposed the LogRobust model [20], which uses the semantic encoding method FastText to extract log semantic information and represent it as a semantic vector for the unstable characteristics of logs, learns semantic information, and detects anomalies through the BiLSTM model, which can categorize log events as semantic approximations when logs are updated, solving the instability of log events caused by log updates, etc. Xia et al., used generative adversarial networks based on LSTM of system logs using permutation event modeling to detect anomalies, called LogGAN [21], to detect log-level anomalies on the basis of patterns. The permutation event modeling mitigates the strong sequential features in LSTM and solves the problem of misordering caused by the late arrival of logs. Generating adversarial networks mitigates the effect of imbalance between normal and abnormal data and improves anomaly detection. Chen et al., argued that it is impractical to build log anomaly detectors to apply to all types of software systems [22], but there are similar aspects among log types of different software systems; they used migration learning to transfer knowledge on the source system to the target system. Haixuan et al., proposed a BERT-based log anomaly detection method LogBERT [23], which learned the patterns of normal log sequences via two self-supervised training tasks and was able to detect anomalies that deviate from normal log sequences. Zhang et al., proposed an unsupervised log sequence anomaly detection network LSADNET based on local information extraction and a global sparse transformer model [24]. Multilayer convolution was applied to capture local correlations between adjacent logs, and a transformer was used to learn global dependencies between distant logs.

The shortcomings of these studies are as follows:

1.  Log formats are constantly changing, yet most methods assumed that log templates do not change. The Word2Vec method was used in most current studies to represent log templates. Word2Vec is a method to convert words into vectors. Using shallow neural network training, words can be converted into vector form. However, Word2Vec is a static word embedding method, and the word vectors of corresponding words are fixed without considering the contextual background, thus extracting no semantic information; it is unable to represent the similarity between different templates with the same or similar semantics, as well as effectively solve the problem of multiple meaning of words.

2.  When the anomaly detection model is applied to a new system, it takes some time to collect enough logs to train the model, and there is a cold-start problem. In a cross-system setup, the log templates of the two may be completely different yet semantically similar. It is not feasible to use the knowledge of one system to fit the anomaly detection model of the other system.

In this paper, we propose the log anomaly detection method LogBD, which uses a pretrained model BERT to extract log semantic information and obtain semantic vectors as features that can better learn the information contained in the logs. Then, a TCN is trained as a feature extractor using a domain adaptation method to extract common features of logs from different systems and build a cross-system log anomaly detection model to detect the source and target anomalies in the system.

## 3. Methodology

### 3.1. Overview

Logs record information about the state of software systems when they are running. They have become an important resource to ensure the reliability and continuity of software systems. However, it is quite difficult to discover system anomalies from massive log data. Based on deep learning and domain adaptation, we propose a domain adaptation-based log anomaly detection method, LogBD, which is outlined in Figure 2.

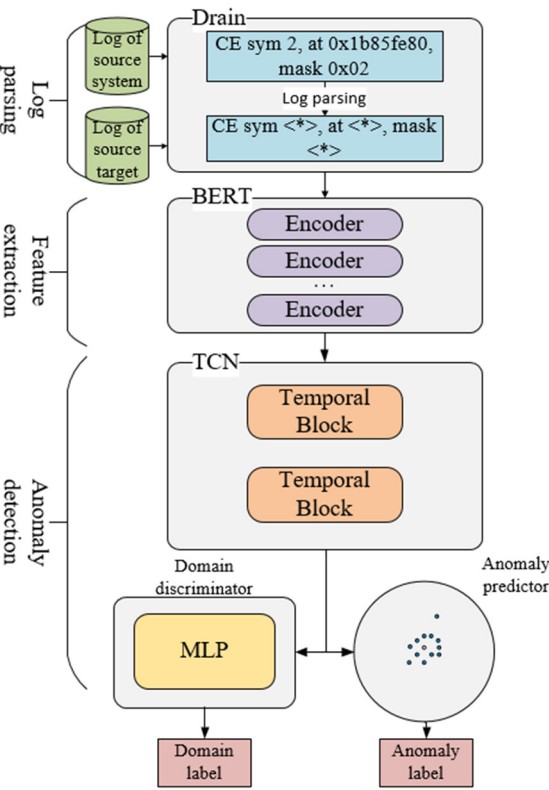

**Figure 2.** The framework of LogBD.

The first step is log parsing, which extracts log contents from unstructured raw logs, and then removes numbers and punctuation marks from logs by regular matching. After comparing some log parsing methods, we use Drain for log parsing, which efficiently converts preprocessed log constants into log templates.

After log parsing, the second step is feature extraction. The log templates are fed into the pretrained model BERT, a bidirectional encoder based on transformer [25] proposed by Google, through which semantic vectors containing semantic information of the log templates are obtained. After representing the log templates as numeric vectors that the neural network can use, the BERT model uses a sliding window method segmentation of the original log sequence into datasets.

The third step is anomaly detection, where the log template sequence after the digital vector representation is fed to a deep learning model for training to learn the patterns of normal log sequences and generate an anomaly detection model based on the distance of the log sequence mapping to the center of the hypersphere space.

### 3.2. Log Parsing Based on Drain

The data required for a general neural network model are structured, and the raw logs usually consist of a large amount of repetitive unstructured data. The task of log parsing is to resolve the raw logs into log templates. Traditional log parsing methods rely on regular expressions, which require manual writing, as well as the need to maintain template updates which cannot be effectively processed for the large amount of log data generated by modern systems. To improve the efficiency and accuracy of anomaly detection, this paper uses Drain to extract relevant information from the logs. For example, if "081109 203518 143 INFO dfs.DataNode$DataXceiver: Received block blk 560063894682806537 of size 67,108,864 from/10.251.194.129" is a raw log from the HDFS log dataset [26], "081109 203518" is the timestamp of the log data, "143" is the process number of the log data, "INFO" is the hazard level of the log data, "dfs.DataNode$DataXceiver" is the component that generated the log data, and "Received block blk 560063894682806537 of size

67,108,864 from/10.251.194.129" is the content of the log data, which records the operation status of this system. The original log contains some irrelevant information, such as IP address, timestamp, and fast ID, which is removed by predefined regular expressions based on domain knowledge, and the content of the log statement is obtained after removing the irrelevant information. The log template is, thus, "Receiving block blk <*> of size <*> from <*>" (Figure 3).

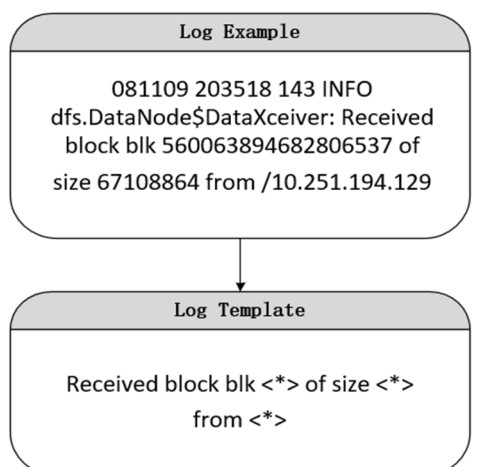

**Figure 3.** Example of log parsing.

### 3.3. Feature Extraction Based on BERT

The log templates are textual data, while the data processed by machine learning and deep learning algorithms are usually fixed-length numerical inputs. The log templates must, therefore, be converted into vectors for computer processing. The word embedding method is used to obtain the semantic vectors.

During the development of natural language processing, word embeddings have undergone an evolution from static to dynamic [27]. Static word embeddings have evolved in two broad stages. In the first stage, mainly sparse and high-dimensional vectors are used to represent words, such as one-hot encoding. Each word of a sentence corresponds to a high-dimensional vector, and all bits in the vector are "0" except for a certain bit which is "1". However, this cannot extract semantic similarity, and the vector dimension of this method is very large as there are many zero values, which wastes storage space. In the second stage, to solve these problems, neural networks are trained using large amounts of text data to generate low-dimensional vectors, with Word2Vec [28] and Glove [29] among the classical approaches. Glove optimizes Word2Vec by considering not only the adjacent local information but also using the co-occurrence matrix to count the global information for better results. Static word embeddings produce vector representations and words in a one-to-one correspondence, which cannot solve the problem of multiple meaning words in log utterances, resulting in incomplete extracted semantic information. Dynamic word embedding methods have emerged, including Embeddings from Language Models (ELMo) [30], Generative Pretraining (GPT) [31], and Bidirectional Encoder Representations from Transformers (BERT) [7]. ELMo extracts the forward and reverse sequence information based on the input text information using a bidirectional LSTM [32] model. The extracted information is used after stitching. The word vector is generated dynamically on the basis of the whole contextual information. GPT uses a one-way transformer as a feature extractor for extracting the above information without considering the reverse contextual information. In contrast, BERT changes the LSTM to a transformer based on ELMo to extract bidirectional context information at the same time, changing from word-level embedding to sentence-level embedding and preserving more semantic information. Firstly, the tokenizer of natural language toolkit (NLTK) package is used to remove the punctuation from the log template, and the * in the template is removed altogether, leaving the token of the

log template to recompose the sentence input to BERT. The embedding layer outputs the vector representation of the sentence by token embedding, position embedding, and segment embedding. The vector representation is then input to the encoding layer, where the encoder module of the two transformers [25] in the encoding layer processes the vector representation. Each word is computed with the attention of the words before and after it to obtain a more accurate contextual representation. After the encoder module is used to achieve the semantic vector encoding of the log sequence, the final pooling layer performs an average pooling operation on the output of all tokens to obtain a fixed size vector as the representation of the whole input sequence. The feature representation process is shown in Figure 4.

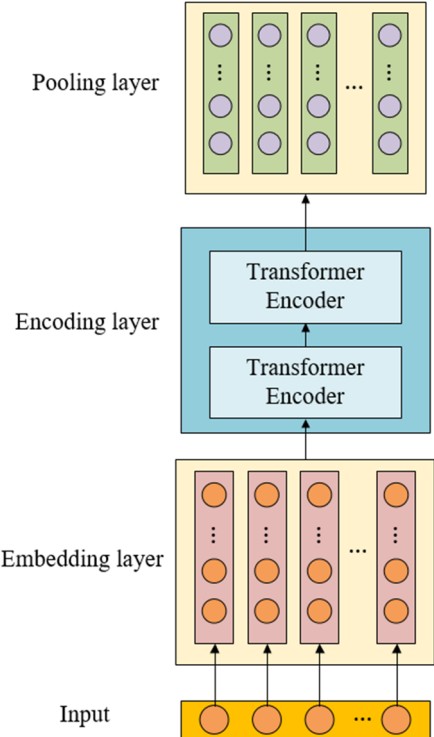

**Figure 4.** Vector representation by BERT.

*3.4. Anomaly Detection Based on Domain Adaptation*

Log anomaly detection is an important step to ensure stable and sustainable operation of software systems, and many machine learning based log anomaly detection methods are proposed to detect anomalous log sequences. However, due to limited anomalous logs, many methods detect anomalous log sequences on the basis of log normal patterns by learning log normal patterns from a large amount of normal log data. Most methods are applied to a system; when deploying to a new system, a large number of logs need to be collected in the new system to retrain the model [22]. For the cold-start problem, several solutions have also been proposed, such as LogTransfer, which assumes that anomalous log sequences in different systems may have similar patterns, and that log message words from different systems usually overlap. Moreover, there are some similarities in the normal workflow of different systems, and migration learning can be used to construct a cross-system anomaly detection model to detect anomalies in the source and target systems. However, LogTransfer requires a large amount of normal and anomaly log data from the source and target systems to construct the anomaly detection model, and it is actually difficult to collect enough log data in a new system [33].

To address the limitations of these previous studies, we propose a domain-adaptive cross-system log anomaly detection method based on LogBD. LogBD establishes an adversarial training framework by domain-adversarial training of neural networks (DANN) [34].

This approach simultaneously makes anomalies in logs and attribution domains, and then achieves the effect of detecting anomalies with confusing data domains to achieve cross-domain (system) log anomaly detection, which requires only a small amount of target system log data to achieve better performance. DANN is a deep learning model for domain adaptation, which achieves feature alignment between source domain systems and target domain systems by adversarial training. It learns feature representations that can effectively classify both source and target domains, and it improves the generalization performance of the model. The network architecture of the DANN model is shown in Figure 5. It contains three parts: feature extractor, anomaly classifier, and domain discriminator.

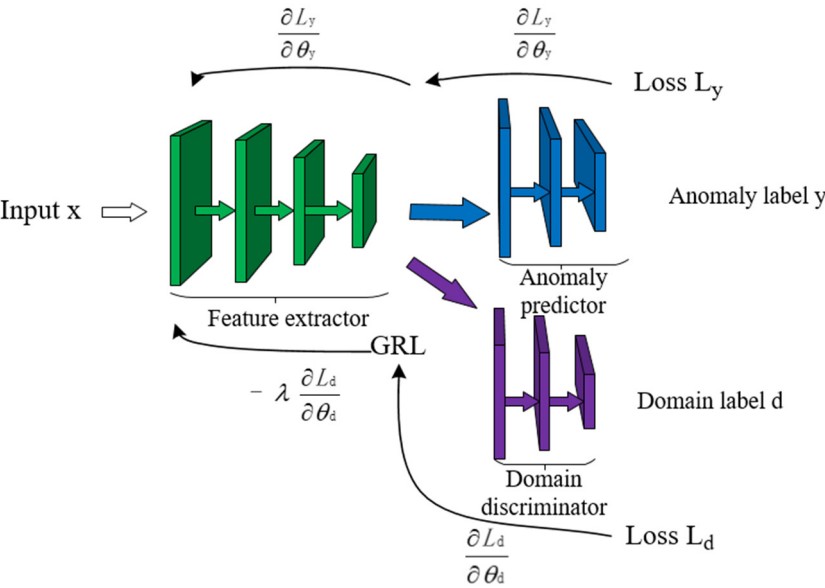

**Figure 5.** The framework of DANN.

### 3.4.1. Feature Extractor

LogBD trains the anomaly detection model using only normal log data. Let D be a dataset consisting of two parts: normal log messages $D^S = \{L_i\}_{i=1}^{Max_S}$ of the source system and a small number of normal log messages $D^T = \{L_i\}_{i=1}^{Max_T}$ of the target system. After generating the log template vector representation using the pretraining model BERT, each log template is converted to a semantic vector representation. A log sequence is represented by $X = \{x_n\}_1^n$, where $x_n$ denotes the vector representation of the n-th log message.

Inspired by the Deep SVDD method, LogBD's feature extractor maps log sequences to hypersphere space [35], which can be processed by neural networks. Deep neural networks are better at extracting the depth information contained in log data than traditional machine learning methods. Since log data carry temporal information, many scholars treat them as time-ordered sequences and use RNNs to learn log the temporal relationships between log entries. Although RNNs establish connections in temporal order with some memorability, they are prone to gradient disappearance or gradient explosion. Schmidhuber proposed an LSTM network based on RNN. Compared with RNN, LSTM introduces a new temporal chain on the basis of RNN. Long-range dependencies can be better captured using the LSTM model because it possesses a gating mechanism to increase or decrease the information of cell states, and ultimately decide which information to use. However, LSTM has some problems; it cannot operate in parallel, only one step can be processed in one operation, and latter steps must wait for the previous steps to be processed before performing the operation. Keeping the intermediate results is very memory-consuming. Therefore, LogBD uses a TCN [8] with better performance and less memory consumption as the feature extractor $G_f$. As shown in Figure 6, a vector representation of log sequences is input to the TCN for encoding and processed to obtain the log sequence mapping v. The i-th sequence

from the source system is mapped as $v_i^S$, and the i-th sequence from the target system is mapped as $v_i^T$.

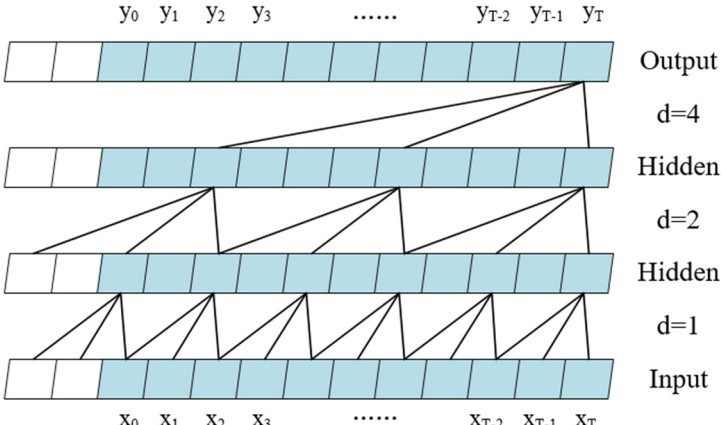

**Figure 6.** Feature extractor processing log sequence representation.

For the feature extractor $G_f$, to have all log sequences mapped into a hypersphere space, we first set the center of the hypersphere to which all log sequences of the source and target systems are mapped as the mean value of all log sequences in the training dataset. The formula is as follows:

$$c = Mean\left(\sum_{i=1}^{Max_\epsilon} v_i^\epsilon\right), \epsilon \in \{S, T\}, \tag{1}$$

where S denotes the source domain or source system, and T denotes the target domain or target system.

Since each log sequence is represented by the feature extractor as $v = G_f(x; \theta_f)$, the distance of the log sequence mapping to the center of the hypersphere sphere is $v_i^\epsilon - c$. In order to make the normal log sequence mapping close to the center of the hypersphere c, the distances of the log sequence mapping to the center of the sphere all need to be close to the minimum, setting the objective function of the feature extractor as follows:

$$\mathcal{L}_{ex} = \sum_{\epsilon \in \{S, T\}} \sum_{i=1}^{Max_\epsilon} \|v_i^\epsilon - c\|^2 + \frac{\lambda}{2} \sum_{l=1}^{L} \|W^l\|_F^2. \tag{2}$$

By minimizing this formula, all normal log sequences can be kept close to the center. The *ex* in the formula represents the feature extractor, the first term in the formula makes all log sequences mapped as close to the center of the sphere as possible, and the second term is regularized.

The loss function of the feature extractor is calculated by the mean square error (MSE), and the feature extractor is optimized to keep the log sequences close to the center of the sphere by minimizing this loss function, which is calculated as follows:

$$loss(v_i, c) = \frac{1}{n} \sum_{i=1}^{n} (v_i - c)^2. \tag{3}$$

### 3.4.2. Adversarial Learning

Although LogBD uses a TCN to map log sequences into the hypersphere space, log sequence mappings from different systems can still be located in different regions of the hypersphere space, rather than clustered around a common hypersphere sphere center c, as shown in Figure 7, where $\mathbb{R}^{128}$ represents a 128-dimensional space.

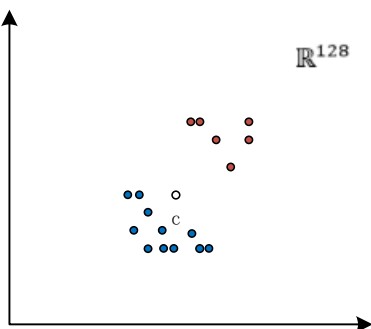

**Figure 7.** Different systems data clustered in different regions.

Therefore, this paper proposes a cross-system data mapping method for adversarial training, which brings log sequence mappings from different systems close to each other. A domain discriminator $G_d$ is set to distinguish whether the log sequence mapping comes from the source or the target system, and the domain discriminator uses a multilayer perceptron containing two 64-dimensional fully connected neural networks. A logistic function is used for prediction, as follows:

$$G_d(v^\epsilon) = \sigma\left(w^T v^\epsilon + b\right), \tag{4}$$

where $\sigma()$ denotes the logistic function, and $w$ and $b$ are trainable parameters.

The loss function of the domain discriminator is calculated by the cross-entropy loss function, which is as follows:

$$loss\left(\hat{d}, d\right) = -\hat{d}[d] + \log\left(\sum_i \exp\left(\hat{d}[i]\right)\right), \tag{5}$$

where $d$ is the true domain label of the log, and $\hat{d}$ is the domain prediction label of the domain discriminator output.

The TCN is trained as a feature extractor, i.e., $v^\epsilon = G_f(X^\epsilon)$, so that it gains the ability to extract features common to both the source and the target systems, making the discriminator unable to distinguish the source system data from the target system data. The adversarial training of the feature extractor and the domain discriminator is achieved by the gradient reversal layer (GRL) [34], where the GRL multiplies the error passed back to the feature extractor by a negative number $(-\lambda)$ from the domain discriminator to achieve the opposite of the training objective and achieve the adversarial effect. The objective function of adversarial training is as follows:

$$\mathcal{L}_{adv} = \underset{G_f \ \ G_d}{minmax}\left(\begin{array}{c} \mathbb{E}_{X^S \sim P_{source}}\left[logG_d\left(G_f(X^S)\right)\right] + \\ \mathbb{E}_{X^T \sim P_{target}}\left[log\left(1 - G_d\left(G_f(X^T)\right)\right)\right] \end{array}\right), \tag{6}$$

where $X^S$ and $X^T$ denote the source log sequence and the target log sequence, respectively.

By minmax optimization training, the log sequence mapping generated by the TCN is able to mislead the domain discriminator, i.e., the distributions of the source and target log sequences are confused. Thus, the objective function training framework can be expressed as the following equation:

$$\mathcal{L} = \mathcal{L}_{ex} + \lambda\mathcal{L}_{adc}, \tag{7}$$

where $\lambda$ is a hyperparameter that is used to balance these two components. Its value is set to 0.1.

During the LogBD training process, an optimizer is needed to update the model parameters. In this paper, we use the Adam optimizer to optimize the model, which maintains the average of the gradient of each parameter, along with the average of the

gradient squared, and then uses these estimates to calculate the adaptive learning rate for each parameter.

### 3.4.3. Anomaly Detection

The anomaly classifier determines whether a log sequence is anomalous according to the distance mapped to the center of the sphere. As shown in Figure 8, the blue normal data points are scattered haphazardly in the hyperspace. By training the feature extractor TCN, the data points are mapped to the hypersphere space, which is centered at c. The TCN is trained to make the normal data points close to the hypersphere sphere center. The anomaly detection is then performed by calculating the anomaly distance threshold using some normal and anomalous data.

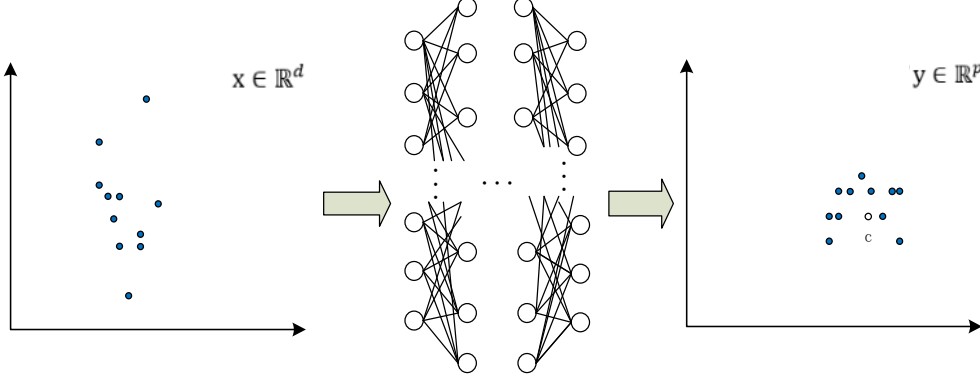

**Figure 8.** Principle of hypersphere anomaly detection; d and p represent dimensions.

After the training phase, the feature extractor gains the ability to map both the source and the target system data to the vicinity of the hypersphere sphere center c, making the anomalous samples of both systems at a large distance from the center of c. To detect anomalies, a threshold $\gamma$ needs to be set as the determination boundary to distinguish between normal and anomalous sequences. The validation set is used to find the optimal thresholds $\gamma^S$ and $\gamma^T$ in the source and target systems, respectively. Samples with a distance greater than $\gamma$ from the center will be marked as abnormal sequences. The validation set is composed of log sequences from any day.

The average values of normal log sequences and abnormal log sequences in the validation set after mapping positions are calculated and denoted as $Mean^{normal}$ and $Mean^{abnormal}$. The threshold is set to the average value of normal sequence positions $Mean^{normal}$, and the abnormal labels of log sequences are given by comparing the distance of log sequences mapped to the center of the hypersphere sphere and the magnitude of the threshold. If the AUC is greater than the current AUC value, the best threshold value is changed to the current threshold setting, and the best AUC is also changed to the current AUC value; otherwise, the best threshold value and the best AUC do not change, and the threshold value increases toward $Mean^{abnormal}$ at the end of judgment to expand the judgment range of normal log sequences. The above operation is cycled 100 times, and the best threshold value $\gamma$ is finally obtained.

## 4. Experiment and Analysis

Firstly, LogBD and log anomaly detection methods PCA, LogCluster, DeepLog, LogAnomaly, LogBERT, and LogTransfer were tested on two datasets and the results were analyzed. The influence of different log vector representation methods on anomaly detection performance was then analyzed, along with the influence of domain adaptation.

*4.1. Experiment Setting*

4.1.1. Dataset

The experimental dataset comes from a large log dataset loghub [26] released by the intelligent operation and maintenance team of the Chinese University of Hong Kong. The total size of the maintained log data set is more than 77 GB. The data were specially collected for log analysis and research. Some of the logs were production data released in previous studies, while others were real data collected from laboratory systems. The logs in the dataset came from various types of systems, including HDFS, BGL, Thunderbird, and OpenSSH. This paper used BGL and Thunderbird datasets for experiments:

- The BlueGene/L (BGL) dataset, which comes from the BlueGene/L supercomputer system of Lawrence Livermore National Labs, contains 4,747,963 logs, including 4,399,494 normal logs and 348,469 abnormal logs.
- The Thunderbird (TB) dataset, from the Thunderbird supercomputer system of Sandia National Labs, contains 3,737,209 logs, including 3,015,004 normal logs and 722,205 abnormal logs.

The statistics of the two datasets are shown in Table 1.

**Table 1.** Dataset information.

| Dataset | Total | Normal | Abnormal |
|---------|-------|--------|----------|
| BGL | 4,747,963 | 4,399,494 | 348,469 |
| TB | 3,737,209 | 3,015,004 | 722,205 |

Simple outlier detection is not needed for group logs because a single abnormal event is considered to be an independent anomaly. However, deep learning models are used to reveal abnormal models of multiple log events, such as changes in event sequences or time log correlations. Therefore, it is necessary to organize log events into groups and then analyze them individually or interrelated. There are two common grouping methods. One is the sliding time window, where a log will appear in multiple windows. The other is a fixed time window, where a log can only appear in one group. In order to ensure the similarity of the experimental environment, a sliding window of 20 was used to cut the log file into a short sequence with a step size of 4.

After sliding window segmentation, the BGL dataset was divided into 1,175,002 log sequences. When BGL was used as the source system, there were 60,000 normal log sequences for training, 1,014,762 normal log sequences for testing, and 100,240 abnormal log sequences for testing. When BGL was used as the target system, 1000 normal log sequences were used for training, 1,073,762 normal log sequences were used for testing, and 100,240 abnormal log sequences were used for testing. The Thunderbird dataset was divided into 1,126,852 log sequences. When Thunderbird was used as the target system, there were 1000 normal log sequences for training, 803,783 normal log sequences for testing, and 322,069 abnormal log sequences for testing. When Thunderbird was used as the source system, there were 60,000 normal logs for training, 744,783 normal log sequences for testing, and 322,069 abnormal log sequences for testing, as shown in Table 2.

**Table 2.** The number of log sequences after sliding window division.

| Dataset | Normal Log Sequence | Abnormal Log Sequence |
|---------|---------------------|------------------------|
| BGL | 1,074,762 | 100,240 |
| TB | 804,783 | 322,069 |

In the training phase, 60,000 normal sequences from the source system and 1000 normal sequences from the target system were mixed as the model training set.

4.1.2. Evaluation Metrics

In order to measure the effectiveness of LogBD in anomaly detection, the $F_1$ value and AUC were used as evaluation indicators. The $F_1$ value combines the precision and recall of the models. A higher $F_1$ value denotes a better performance of the classification model. Precision represents the percentage of real anomalies in all detected anomalies, recall represents the percentage of anomalies detected in the dataset, and the $F_1$ value is calculated as follows:

$$F_1 = 2\frac{Precision \times Recall}{Precision + Recall}. \tag{8}$$

The formulas of precision and recall are as follows:

$$Precision = \frac{TP}{TP + FP}. \tag{9}$$

$$Recall = \frac{TP}{TP + FN}. \tag{10}$$

The AUC is the area under the ROC curve. The closer the area is to 1, the better the model effect is. It is mainly used to measure the performance of a binary classification model. The AUC calculation formula is as follows:

$$AUC = \frac{\Sigma_{ins_i \in positiveclass} rank_{ins_i} - \frac{M*(M+1)}{2}}{M \times N}, \tag{11}$$

where $rank_{ins_i}$ is the ordinal number of the i-th sample, M and N are the number of positive and negative samples, respectively, and $\Sigma_{ins_i \in positiveclass}$ is the sum of the ordinal numbers of all positive samples.

The value range of the $F_1$ value is 0 to 1. A higher value denotes a better performance of the classifier and a better model effect, which can comprehensively describe the detection performance of the model. The value interval of AUC is 0.5 to 1. A higher value denotes a better performance of the model. AUC is not sensitive to abnormal thresholds.

4.1.3. Baselines

LogBD was compared with the following log anomaly detection methods to evaluate the performance of LogBD:

- PCA (principal component analysis) is based on a dimensionality reduction anomaly detection method, combined with term weighting technology in the field of data retrieval.
- logCluster, based on a similarity clustering method, can automatically find similar log patterns and aggregate them into clusters to achieve automatic identification of anomalies.
- DeepLog, an anomaly detection model based on log templates, uses LSTM to model log data, automatically learns abnormal behavior patterns, and compares predicted logs with real logs for anomaly detection.
- LogAnomaly integrates the log template by analyzing the synonyms in the log, encodes the log template into a semantic vector, and inputs it into the LSTM model for anomaly detection.
- LogBERT, a BERT-based method LogBERT, constructs two self-supervised training tasks to learn the pattern of normal log sequences and detect anomalies that deviate from the normal model.
- LogTransfer uses transfer learning to achieve cross-system anomaly detection, utilizes labeled normal and abnormal data of two systems to train the model, and shares a fully connected network that classifies logs.

### 4.1.4. Implementation

The model designed in this paper was built on a cloud server, The configuration used in the experiment was CPU Intel (R) Xeon (R) Platinum 8255C, GPU RTX 2080Ti, memory 43 GB, Pytorch 1.7.1.

### 4.2. Results and Analysis

Since PCA, LogCluster, DeepLog, LogAnomaly, and LogBERT are unsupervised models and are not designed for cross-system detection, these models were evaluated in two cases; that is, the training dataset used or did not use the samples of the target system, represented by W/O. Firstly, the Thunderbird dataset was used as the source system, and the BGL dataset was used as the target system. This situation is abbreviated as TB-BGL. The $F_1$ value of 0.880 and the AUC value of 0.880 were obtained on the source system Thunderbird, and the $F_1$ value of 0.938 and the AUC value of 0.973 were obtained on the target system BGL. Then, the BGL dataset was used as the source system and the Thunderbird dataset was used as the target system. This case is abbreviated as BGL-TB. The $F_1$ value of 0.933 and the AUC value of 0.978 were obtained on the source system BGL, and the $F_1$ value of 0.841 and the AUC value of 0.854 were obtained on the target system Thunderbird.

Tables 3 and 4 show the experimental results of the six methods except LogTransfer in the case of TB-BGL and BGL-TB. W/O represents the training set using or not using the target system. Firstly, PCA, LogCluster, DeepLog, LogAnomaly, and LogBERT do not use domain adaptation. For Thunderbird as the source system and BGL as the target system (TB→BGL) or BGL as the source system and Thunderbird as the target system (BGL→TB), these five methods could produce good $F_1$ values and AUC values on the source system even if they did not use the sample training of the target system. When the training dataset used samples from the target system, they could obtain better $F_1$ and AUC values on the target system, but worse values on the source system, indicating that these five methods do not have cross-system adaptive ability. Using a mixed log sequence of the source system and the target system would only make the training data distribution diverse, whereby the detection model would confuse the distribution of the samples, resulting in poor model detection. Compared with these five methods, LogBD achieved better results in any scenario.

**Table 3.** The experimental results of six methods in the case of TB-BGL.

| Method | Source | | Target | |
|---|---|---|---|---|
| | $F_1$ | AUC | $F_1$ | AUC |
| PCA W/O | 0.689/0.760 | 0.698/0.779 | 0.577/0.229 | 0.773/0.658 |
| LogClusterW/O | 0.708/0.724 | 0.688/0.716 | 0.597/0.223 | 0.686/0.500 |
| DeepLog W/O | 0.757/0.790 | 0.701/0.777 | 0.627/0.223 | 0.843/0.500 |
| LogAnomaly W/O | 0.793/0.813 | 0.754/0.791 | 0.697/0.287 | 0.859/0.647 |
| LogBERT W/O | 0.821/0.847 | 0.805/0.857 | 0.732/0.349 | 0.851/0.682 |
| LogBD (ours) | **0.880** | **0.880** | **0.938** | **0.973** |

**Table 4.** The experimental results of six methods in the case of BGL-TB.

| Method | Source | | Target | |
|---|---|---|---|---|
| | $F_1$ | AUC | $F_1$ | AUC |
| PCA W/O | 0.322/0.642 | 0.587/0.816 | 0.731/0.558 | 0.776/0.504 |
| LogClusterW/O | 0.530/0.713 | 0.746/0.829 | 0.677/0.559 | 0.716/0.504 |
| DeepLog W/O | 0.834/0.878 | 0.824/0.867 | 0.720/0.656 | 0.779/0.600 |
| LogAnomaly W/O | 0.859/0.905 | 0.901/0.936 | 0.784/0.643 | 0.809/0.691 |
| LogBERT W/O | 0.882/0.926 | 0.915/0.954 | 0.809/0.709 | 0.828/0.733 |
| LogBD (ours) | **0.933** | **0.978** | **0.841** | **0.854** |

On the source system, the detection performance of LogBD was still better than the five models, which indicates that LogBD captured the pain points of log anomaly detection, i.e., the accuracy of log template analysis, the use of log semantic information, and the method of anomaly detection. It can also be seen that the performance of deep learning methods was better than that of machine learning methods, indicating that the machine learning model only used the log template count vector as the input feature, without considering the log content itself. Machine learning can detect the abnormal information in the log to a certain extent, but it cannot achieve good accuracy and coverage of anomaly detection. For example, PCA is based on the log template index for anomaly detection, only retains the main features of the original data, and loses a lot of key information, making it difficult to learn features from the sparse count matrix. LogCluster is based on clustering for log anomaly detection, but it cannot play a good role in the face of complex log structure, nor can it fully learn the features in the log, and the detection effect is not good. DeepLog regards the log sequence as a digital sequence and replaces the log template with a number. It not only uses the log parameter features but also integrates the log sequence features. However, it does not extract the semantic information in the log template, and it can easily treat the log sequence that has not appeared in the training data as an exception, resulting in lower accuracy and more false alarms. Compared with the machine learning methods PCA and LogCluster, it has a more obvious improvement. The LogAnomaly method uses the semantic and grammatical information of the log template, and then introduces Template2Vec for the synonyms in the log. It uses the word vector weighted average to obtain the vector representation, which makes some improvements on the basis of DeepLog. However, it does not consider the problem of polysemous words, only considers the representation of a single word vector, and does not consider the context information; additionally, the learned feature information is not comprehensive enough. LogBERT uses BERT to capture the pattern of normal log sequence, and uses two self-supervised task training models, mask log template prediction and hypersphere minimization. LogBERT uses the hypersphere objective function as in LogBD, but the performance is not as good as LogBD, because LogBD uses domain adaptation to obtain more data with the same characteristics for training.

LogTransfer is a supervised transfer learning method that uses normal and abnormal labeled data of the source system and the target system to train a cross-system anomaly detection model. LogTransfer achieves good performance when sufficient tag data are available. In this experiment, we tested how many labeled abnormal samples are needed to train LogTransfer for it to have similar performance to LogBD. When using 100 abnormal sequences from the source system and 10 abnormal sequences from the target system to train LogTransfer, LogTransfer could achieve the best performance on the source system. The detection results in the two scenarios are shown in Tables 5 and 6.

**Table 5.** The case of TB-BGL.

| Method | Source | | Target | |
| --- | --- | --- | --- | --- |
| | $F_1$ | AUC | $F_1$ | AUC |
| LogTransfer | **0.981** | **0.975** | 0.788 | 0.833 |
| LogBD (ours) | 0.880 | 0.880 | **0.938** | **0.973** |

**Table 6.** The case of BGL-TB.

| Method | Source | | Target | |
| --- | --- | --- | --- | --- |
| | $F_1$ | AUC | $F_1$ | AUC |
| LogTransfer | **0.971** | 0.972 | 0.792 | 0.828 |
| LogBD (ours) | 0.933 | **0.978** | **0.841** | **0.854** |

For the TB→BGL scenario, training LogTransfer with 10 abnormal sequences of the target system was not enough to be better than LogBD. It was better than LogBD on the source system, but worse on the target system. For the BGL-TB scenario, the performance was comparable on the source system and lower than LogBD on the target system. Therefore, LogBD can provide good performance using only normal data when it is difficult to obtain labeled abnormal samples.

Unlike previous methods, LogBD uses BERT to extract the semantic information of log messages, and uses semantic vectors to represent log templates, rather than the log templates Word2Vec and Glove used in previous methods. This paper compared the model performance using four log template representation methods. The detection results generated by different log template representation methods in two scenarios are shown in Tables 7 and 8. We found that the performance was greatly improved when BERT was used for log template representation. This may be because the method based on the log template ID is to number the log template and represent it by number. This method regards the log template as a number; it does not consider the semantic information contained in the log template. Word2Vec and Glove word vectors are fixed by looking up the dictionary and taking the corresponding word vectors. They cannot dynamically adjust the word vectors according to different context contexts, losing the integrity of the log template semantics. According to the context of the input sentence, BERT returns sentence-level word vectors through model calculation in the model network. Due to the different context of the input context, the returned word vectors are different in order to distinguish polysemous words. Compared with the first three, BERT can learn the deep semantics of sentences and capture the similarity between different log statements.

**Table 7.** Detection results generated by different log template representation methods in the case of TB-BGL.

| Method | Source | | Target | |
|---|---|---|---|---|
| | $F_1$ | AUC | $F_1$ | AUC |
| Log Template ID | 0.725 | 0.733 | 0.784 | 0.776 |
| Word2Vec | 0.788 | 0.797 | 0.845 | 0.909 |
| Glove | 0.832 | 0.857 | 0.897 | 0.918 |
| BERT (ours) | **0.880** | **0.880** | **0.938** | **0.973** |

**Table 8.** Detection results generated by different log template representation methods in the case of BGL-TB.

| Method | Source | | Target | |
|---|---|---|---|---|
| | $F_1$ | AUC | $F_1$ | AUC |
| Log Template ID | 0.781 | 0.799 | 0.679 | 0.691 |
| Word2Vec | 0.813 | 0.854 | 0.728 | 0.774 |
| Glove | 0.885 | 0.862 | 0.792 | 0.828 |
| BERT (ours) | **0.933** | **0.978** | **0.841** | **0.854** |

The cross-system log anomaly detection model LogBD proposed in this paper uses the domain adaptation method in transfer learning and achieves excellent performance. In order to prove the effectiveness of the domain adaptation method, a set of comparative experiments was carried out to compare the performance of the model without using the domain adaptation method and using the domain adaptation method. The impact of domain adaptation methods on model performance was evaluated.

The detection results using or not using the domain adaptation method in the two scenarios are shown in Tables 9 and 10. "Without" indicates that the domain adaptation method was not used, while "with" indicates that the domain adaptation method was used. Through observation, it can be seen that the domain adaptive method could greatly

improve the performance of the anomaly detection model, because the anomaly detection model could learn the similarities between the two system log data, thereby detecting the anomalies of the two systems.

**Table 9.** TB-BGL scenario using or not using domain adaptation method.

| Domain Adaptation | Source | | Target | |
|---|---|---|---|---|
| | $F_1$ | AUC | $F_1$ | AUC |
| Without | 0.794 | 0.807 | 0.467 | 0.418 |
| With (ours) | **0.880** | **0.880** | **0.938** | **0.973** |

**Table 10.** BGL-TB scenario using or not using domain adaptation method.

| Domain Adaptation | Source | | Target | |
|---|---|---|---|---|
| | $F_1$ | AUC | $F_1$ | AUC |
| Without | 0.647 | 0.872 | 0.592 | 0.628 |
| With (ours) | **0.933** | **0.978** | **0.841** | **0.854** |

This experiment also further evaluated the performance of LogBD when training with different numbers of target system normal log sequences. The experimental results are shown in Figure 9.

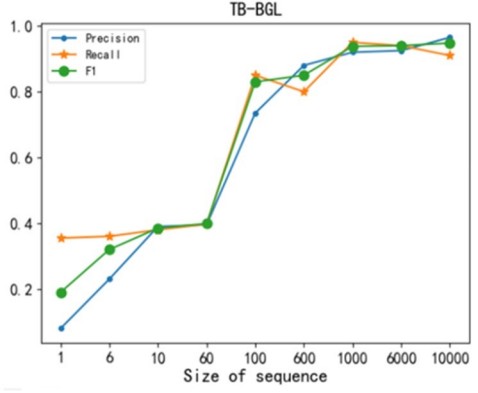 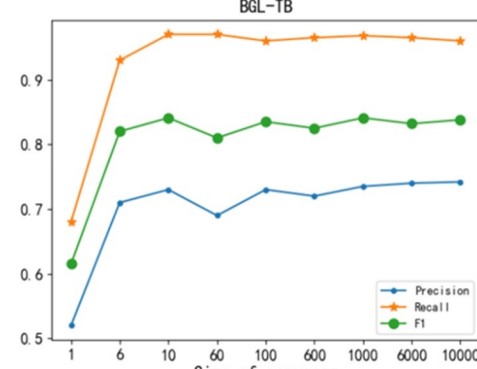

**Figure 9.** (**Left**) TB-BGL; (**right**) BGL-TB.

It can be observed that LogBD could achieve high anomaly detection performance for the target system by using a small number of normal sequences in the target system. In the TB→BGL scenario, using about 100 normal log sequences in BGL could achieve good performance. For the BGL→TB scenario, only using 10 normal log sequences in Thunderbird was enough to obtain good performance. When the number of log sequences increased, the performance continued to improve. In general, even if the new online system deployment time is short, it is easy to obtain normal log data from the target system; thus, LogBD has strong feasibility and accuracy in detecting anomalies in the new system.

## 5. Conclusions

Aiming at the problems existing in the current log anomaly detection methods, this paper proposed a cross-system log anomaly detection method LogBD based on domain adaptation. This method establishes an adversarial training framework based on the adversarial domain adaptive network, uses the pretraining model BERT to extract the semantic information of the log statement, uses the TCN to learn the order relationship of the log sequence, and maps the log sequence from the two systems to the hypersphere space, so that they are close to its center. On the basis of the distance between the log sequence and the center of the sphere, the anomaly is judged, and the distribution of the

domain is confused. At the same time, a good anomaly detection effect is obtained. Related experiments of LogBD were designed and carried out. The experiments showed that the method achieved good results in cross-system log anomaly detection.

In this paper, the log data were divided into log sequences through the sliding window mechanism; only the log template sequence was used for anomaly detection, whereas the parameter information in the log message was not used. The next research direction is to establish new anomaly detection schemes for some important parameters, such as delay anomaly for log interval. Although the log anomaly detection model in this paper considers cross-system anomaly detection, it can only realize anomaly detection on two systems, whereas it cannot be deployed on multiple different systems. The next research direction will be to continue to study multisystem log anomaly detection in the field of domain adaptation.

**Author Contributions:** Conceptualization, L.D. and S.L.; methodology, L.D.; software, L.D.; validation, L.D., H.X. and W.W.; formal analysis, L.D.; investigation, L.D.; resources, L.D.; data curation, L.D.; writing—original draft preparation, L.D.; writing—review and editing, L.D.; visualization, L.D.; supervision, L.D.; project administration, S.L.; funding acquisition, S.L. All authors have read and agreed to the published version of the manuscript.

**Funding:** This research was funded by the National Nature Science Foundation of China, grant number 61762085, and the Autonomous Region Science and Technology Program of Xinjiang, grant 2020A02001-1.

**Institutional Review Board Statement:** Not applicable.

**Informed Consent Statement:** Not applicable.

**Data Availability Statement:** The data presented in this study are openly available from [26].

**Acknowledgments:** The authors thank NSFC for funding this research, as well as the anonymous reviewers for their contribution to this paper.

**Conflicts of Interest:** The authors declare no conflict of interest.

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
