# Peer review of "LogBD: A Log Anomaly Detection Method Based on Pretrained Models and Domain Adaptation"

_applsci, doi:10.3390/app13137739_

Round 1

Reviewer 1 Report

The paper presents LogBD, a novel log anomaly detection method that leverages pre-trained models and domain adaptation techniques to improve the accuracy of anomaly detection in log data. 

suggestion for improvement 

1. Abstract needs to revise, as mention the actual contribution of authors

2. The paper briefly mentions the scalability of the LogBD method but does not provide an in-depth analysis or empirical results regarding its scalability to larger log datasets. 

3. Discussion on Adversarial Learning: The authors mention using them but do not provide sufficient details or a thorough discussion on their selection and impact. It would strengthen the paper's technical depth by providing insights into the rationale behind the chosen hypersphere space and their effects on the model's performance also explain figure 7.

4. In result Section, confusion matrix is missing. Although Table 7 & 8 shows the results but graphical representation is must.  

The quality of English language in this paper is good

Author Response

Dear experts, thank you very much for taking the time to review my paper, I have made amendments to your comments : 1.Modified the abstract. 2.The in-depth analysis or empirical results of the scalability of large log data sets are not provided because it is difficult to find some log data with different contents but similar semantics, which may not be suitable for domain adaptation. 3.The domain adaptation used in the paper is based on DANN, which comes from the paper Domain-Adversarial Training of Neural Networks. 4.This form is mostly used in the field of log anomaly detection, such as LogTransfer, and does not use confusion matrix.

Reviewer 2 Report

My comments are in the attached file.

The English needs major revision.

Author Response

Dear experts, thank you very much for taking the time to review my paper, I have made amendments to your comments

Reviewer 3 Report

The paper is devoted to a technique to find anomalies using the log-based data. The authors use BERT model to obtain semantical information from logs as well as a TCN network to detect anomalies. 

The reviewer can state that the anomaly detection methods are very useful now because they do not require a domain-oriented mathematical model fro the system. The paper tries to provide a general solution to it. 

In principle, the article gives us an overall view on the problem of anomaly detection, but it lacks for some fundamential explanations. 

First of all, the definition of a anomaly is not given. What is an anomaly in a mathematical point of view?

Secondly, there are two many different cases: (1) anomaly in data (no context), (2) anomaly in data series (there is context or state). The first one can be detected using an ordinary neural network, the latter using a recursive NN or different analogues to take the context in mind. In the paper, it is very unclear which case is used. Seems to be the last one (as they write about the sliding window, but this concept again is not explained in the appropriate images). 

The authors write too much about log parsing but it is only a technical problem. Instead, they should explain how the BERT and TCN work in details. (Use a simple case and explain it step-by-step). 

How the authors filter point anomalies (i.e. problems with data measuring)? In general, some wrong points can be skipped if they do not appear again in the future. 

Also, providing F-score for the evaluation is not sufficient: in the article, the authors should compare their method against others (see the works on Anomaly Benchmarks, for example, from Jeff Hawkins team). 

It is very strange that the authors compare the results with clustering methods. Seems, the anomaly is more connected with time series analysis and not clustering, please prove such method selection. 

Some found anomaly cases from the given datasets should be well studied by hand and explained to the readers. 

Minor problems: 

- Fig. 3 is a low-quality screenshot from Word with red syntax highlighting, inacceptable in the article! 

- Relate Work

- Log anomaly detection Log anomaly detection (line 110).

All the text should be rechecked! 

Considering the above, the reviewer thinks that the article is an initial stage for publishing and should be seriously revised.

Should be revised, the phrases should be splitted; sometimes, the phases are not correct since the text is not well checked. 

Author Response

尊敬的各位专家,非常感谢您抽出宝贵时间审阅我的论文,我对您的评论进行了修改: 1.日志异常类型可以分为异常值异常、序列异常、频率异常和统计异常,我们研究序列异常,类似于你提到的第二种情况,将日志序列视为一个时间序列,而不考虑过滤点异常。2.图片和文字经过重新检查。

Reviewer 4 Report

The authors have introduced a novel log anomaly detection method called LogBD. This method leverages the pretraining model BERT to capture the semantic information of logs and addresses challenges arising from multiple-meaning words and log statement updates. By utilizing domain adaptation techniques, LogBD demonstrates promising capabilities in detecting anomalies within log data.

- The subsection titled "2.2. Feature Extraction" is situated within the "2. Related Work" section; however, the authors did not reference any prior work specifically related to feature extraction. To address this, the authors have the option to either include a separate technical background section or introduce and discuss the importance of feature extraction within the relevant context as needed throughout the paper.

- To enhance the presentation of the "anomaly detection" related work, it is suggested that the authors consider including a table that provides a comprehensive summary of the surveyed papers. This table can incorporate the three mentioned categories: rule-based, graph-based, and machine learning-based detection methods. By comparing the proposed method with existing approaches in the table, readers will gain a clearer understanding of the unique contributions and advantages of the authors' approach.

- To enhance clarity and improve the understanding of the paper, it is recommended to rephrase and articulate the titles of the subsections included in the second and third sections. This will help provide a more accurate and concise representation of the content covered in each subsection, making it easier for readers to grasp the main ideas and topics discussed within the sections.

- Add clear legend and axis titles to Figure 7 and Figure 8.

It is important to revise the English language. The text contains many grammatical mistakes and lacks clarity. 

Author Response

Dear experts, thank you very much for taking the time to review my paper, I have made amendments to your comments : For feature extraction, its technical background has been introduced in 2.2 ; the unique contributions and advantages of the method are given in 2.3. The subheading of the third section has been modified ; figure 7 and Figure 8 represent a multi-dimensional space with no coordinate axis.
